CXCL10-based gene cluster model serves as a potential diagnostic biomarker for premature ovarian failure

Qin Ying 1 2 ogyingqin@126.com
Wen Canliang 1
Wu Huijiao 2
1 Department of Obstetrics and Gynecology, Guangzhou Women and Children’s Medical Center , Guangzhou , China
2 Reproductive Medicine Center, Guangzhou Women and Children’s Medical Center , Guangzhou , China
Dong Peixin
Electronic publication date: 2023 Dec 13
Publication date: 2023
Volume: 11
Electronic Location ID: e16659
Received 2023 Sep 21; Accepted 2023 Nov 21
Copyright: © 2023 Qin et al.
Copyright year: 2023
Copyright holder: Qin et al.
License: This is an open access article distributed under the terms of the Creative Commons Attribution License, which permits unrestricted use, distribution, reproduction and adaptation in any medium and for any purpose provided that it is properly attributed. For attribution, the original author(s), title, publication source (PeerJ) and either DOI or URL of the article must be cited.
License URL: https://creativecommons.org/licenses/by/4.0/

Keywords: Premature ovarian failure, CXCL10, Diagnostic biomarker, PPAR signaling pathway

Funding: Basic Research Project of Guangzhou Municipal Science and Technology Bureau 202102010293 Guangzhou Women and Children’s Medical Center 2019BS034 This work was supported by the Basic Research Project of Guangzhou Municipal Science and Technology Bureau (No. 202102010293) and the Doctor’s Fund of Guangzhou Women and Children’s Medical Center (No. 2019BS034). The funders had no role in study design, data collection and analysis, decision to publish, or preparation of the manuscript.

==============================
Objective

Premature ovarian failure (POF) is a disease with high clinical heterogeneity. Subsequently, its diagnosis is challenging. CXCL10 which is a small signaling protein involved in immune response and inflammation may have diagnostic potential in detection of premature ovarian insufficiency. Therefore, this study aimed to investigate CXCL10 based diagnostic biomarkers for POF.

Methods

Transcriptome data for POF was obtained from the Gene Expression Omnibus (GEO) database (GSE39501). Principal component analysis (PCA) assessed CXCL10 expression in patients with POF. The receiver operating characteristic (ROC) curve, analyzed using PlotROC, demonstrated the diagnostic potential of CXCL10 and CXCL10-based models for POF. Differentially expressed genes (DEGs) in the control group of POF were identified using DEbylimma. PlotVenn was used to determine the overlap between the POF-control group and the high-/low-expression CXCL10 groups. QuadrantPlot was employed to detect CXCL10-dysregulated genes in POF. Gene Ontology (GO), Kyoto Encyclopedia of Genes and Genomes (KEGG), and Gene Set Enrichment Analysis (GSEA) were conducted on DEGs using RunMulti Group cluster Profiler. A POF model was induced with cisplatin (DDP) using KGN cells. RT-qPCR and Western blot were used to measure the expression of CXCL10, apoptosis-related proteins, and peroxisome proliferator–activated receptor (PPAR) signaling pathway-related proteins in this model, following siRNA-mediated silencing of CXCL10. Flow cytometry was employed to assess the apoptosis of KGN cells after CXCL10 downregulation.

Results

The expression of CXCL10 is dysregulated in POF, and it shows promising diagnostic potential for POF, as evidenced by an area under the curve value of 1. In POF, we found 3,362 up-regulated and 3,969 down-regulated DEGs compared to healthy controls, while the high- and low-expression groups of POF (comprising samples above and below the median CXCL10 expression) exhibited 1,304 up-regulated and 1,315 down-regulated DEGs. Among these, 786 DEGs consistently displayed dysregulation in POF due to CXCL10 influence. Enrichment analysis indicated that the PPAR signaling pathway was activated by CXCL10 in POF. The CXCL10-based model (including CXCL10, Itga2, and Raf1) holds potential as a diagnostic biomarker for POF. Additionally, in the DDP-induced KGN cell model, interfering with CXCL10 expression promoted the secretion of estradiol, and reduced apoptosis. Furthermore, CXCL10 silencing led to decreased expression levels of PPARβ and long-chain acyl-CoA synthetase 1 compared to the Si-NC group. These results suggest that CXCL10 influences the progression of POF through the PPAR signaling pathway.

Conclusion

The CXCL10-based model, demonstrating perfect diagnostic accuracy for POF and comprising CXCL10, Itga2, and Raf1, holds potential as a valuable diagnostic biomarker. Thus, the expression levels of these genes may collectively provide valuable diagnostic information for POF.

Introduction

Premature ovarian failure (POF) is a disease with high clinical heterogeneity (Luisi et al., 2015), and its incidence is rising with rapid socioeconomic development. Hormone supplement therapy is the primary treatment method for POF, alleviating clinical symptoms but failing to improve ovarian function (Chon, Umair & Yoon, 2021). Therefore, establishing a safe and effective diagnosis for patients with POF has gained significant attention.

Transcriptome-based prognostic signatures entail specific gene expression patterns predicting disease outcome and prognosis (Ahluwalia, Kolhe & Gahlay, 2021; Lemij et al., 2023). They aid in identifying genes or groups of genes whose expression levels are associated with specific clinical outcomes, such as disease progression, survival rates, or treatment response. These gene expression signatures can provide valuable information for personalized medicine, treatment decision-making, and patient management (Ma et al., 2022), shedding light on the molecular mechanisms underlying disease progression and facilitating tailored treatment (Zhou et al., 2022).

Several genes, including GDF9, BMP15, NGF, FANCM, STAG3, FSHR, NRIP1, XPO1, and MACF1, have relevance in the genetic diagnosis, research, and clinical practice of POF (Dixit et al., 2010; Pouresmaeili & Fazeli, 2014). Jaillard et al. (2020) identified GDF9, FANCM, STAG3, and FSHR as being involved in POI pathogenesis, and proposed novel associated candidate genes, NRIP1, XPO1, and MACF1, to be further investigated. Fassnacht et al. (2006) conducted a diagnostic analysis in 101 patients with POF, investigating the major candidate genes (DAZL, DBX, FOXL2, INHalpha, GDF9, USP9X) that contribute to POF development. While several transcriptome analyses in POF have been performed (Kuang et al., 2014; Li et al., 2019), a specific molecular diagnostic marker for POF remains elusive.

CXCL10, a small signaling protein, is produced by various cell types, including immune cells, endothelial cells, and fibroblasts, in response to immune and inflammatory signals such as interferon-gamma (IFN-γ).

CXCL10 is involved in immune response, host defense against viral and bacterial infections, and several inflammatory and autoimmune diseases (Karin & Razon, 2018; Nakagome & Nagata, 2022). In autoimmune diseases, CXCL10 contributes to tissue damage by promoting the infiltration of immune cells into affected tissues (Ghafouri-Fard et al., 2021; Tokunaga et al., 2018). Additionally, it may have diagnostic potential in the early detection of premature ovarian insufficiency (POI) and a role in ovarian fibrosis (Wang & Sun, 2022). However, the exact function and clinical significance of CXCL10 in POF remains unclear.

Gene-based diagnostic models have the potential to revolutionize disease diagnosis, personalized medicine, and patient management, offering more accurate and targeted approaches to healthcare (Han et al., 2022). They offer the possibility of early detection, improved treatment selection, and better patient outcomes (Thaker et al., 2019). POF results from various genetic causes, and while a single definitive diagnostic gene marker remains elusive, gene-based markers can aid in some cases.

This study explores CXCL10-based diagnostic biomarkers for POF, investigating its expression, global regulation, and related signaling pathways. Furthermore, we developed and validated a highly accurate CXCL10-based model for diagnosing POF.

Materials and Methods

Data collection

Transcriptome data associated with POF was obtained from the Gene Expression Omnibus (GEO) database (dataset: GSE39501). This dataset comprised gene expression data from three POF mouse samples (cases, including GSM970254, GSM970255, and GSM970256) and three wild-type mouse samples (controls, including GSM970248, GSM970249, and GSM970250). The POF mouse samples were ovaries obtained from C57BL/6 mice with induced POF, while the control mouse samples were ovaries obtained from wild-type C57BL/6 mice. The three POF mouse samples were further divided into high and low CXCL10 expression groups using as a threshold the median expression level of CXCL10 (4.059386919). The data underwent standardization using RunNorm, a tool developed based on the Limma tool (Ritchie et al., 2015).

Principal component analysis (PCA)

PCA is a standard technique for reducing dimensionality in multivariate statistical analysis, emphasizing the most variance-contributing dataset features. By using PCA, we reduced dataset dimensionality and created a two-dimensional scatter map for an initial sample distribution overview (Li et al., 2022). PCA was performed using the PCA expression plot feature in RStudio. The transcriptional levels of CXCL10 in POF and control groups were visualized using ViolinPlot, which generates violin plots based on the result file and grouping information output with the RunNorm application.

Receiver operating characteristic (ROC) curve

The ROC curve is a common clinical practice tool that can be used to evaluate the performance of classifiers. An area under the curve (AUC) value exceeding 0.5 and approaching 1, indicates better diagnostic accuracy. AUC values ranging from 0.5 to 0.7 indicate low accuracy, values ranging from 0.7 to 0.9 indicate moderate accuracy, and values exceeding 0.9 indicate high accuracy. ROC analysis was performed to assess the optimum cutoff value, sensitivity, specificity, and AUC with a 95% confidence interval (CI). The ROC curve of CXCL10 was plotted and analyzed using the PlotROC application, based on the R language package pROC (Sachs, 2017).

The Least Absolute Shrinkage and Selection Operator (LASSO) regression is a commonly used data mining method in machine learning known for variable selection and complexity adjustment within generalized linear models to mitigate multicollinearity in regression analysis. We utilized Baiyin Cloud to implement RunLASSO for selecting feature genes impacted by CXCL10 in POF with diagnostic potential. This application, based on the R language LASSO package, establishes a LASSO model (Friedman, Hastie & Tibshirani, 2010) while generating Lambda plots, LASSO model diagrams, and ROC curves. Transcriptional expression of model genes was depicted using PlotBox showcasing them in a box plot.

Gene ontology (GO), Kyoto Encyclopedia of Genes and Genomes (KEGG), and Gene Set Enrichment Analysis (GSEA) pathway analysis

The “clusterProfiler” package was used to perform GO, KEGG, and GSEA pathway analyses of the differentially expressed genes (DEGs) to annotate biological functions and enriched pathways. Enrichment results with false discovery rate (FDR) <0.05 were considered significant. POF dysregulation genes influenced by CXCL10 underwent enrichment analysis using RunMutiGroupclusterProfiler, an application developed based on the clusterProfiler package in R language (Yu et al., 2012), for GO and KEGG. The significance of the enrichment results was based on a p-value < 0.05. The enrichment results for biological processes (BP) or KEGG pathways were visualized using the PlotClusterBubble application on Baiyin Cloud.

Cell proliferation assay

The cell counting kit-8 (CCK-8) method was used for assessing cell proliferation. KGN cells were cultured in a 96-well plate for 24 or 48 h. Then, 10 μL of CCK-8 medium (HY-K0301; MCE, Concord, CA, USA) was added to each well. After a 1-h incubation at 37 °C with 5% CO2, the optical density (OD) at 450 nm was measured. Each measurement was conducted in triplicate.

Cell culture and transfection

Human ovarian granulosa cells (KGN cells) were obtained from the American Type Culture Collection (ATCC) and cultured in RPMI-1640/F12 medium (Gibco, Waltham, MA, USA) supplemented with 10% fetal bovine serum (FBS; Gibco, Waltham, MA, USA), 100 U/mL penicillin, and 100 mg/mL streptomycin. The cells were grown at 37 °C with 5% CO2. CXCL10 short interfering RNAs (siRNAs) were synthesized by Shanghai Gene Pharma Co., Ltd. (Shanghai, China). Lipfectmine3000 (Thermo Fisher, Waltham, MA, USA) was used to transfect the siRNAs at a final concentration of 50 nM, following the manufacturer’s instructions.

For cisplatin (DDP) treatment (Wang et al., 2022), the KGN cells were treated with different concentrations of substance (0, 10, 20, 40, and 80 μg/mL) for 24 or 48 h. The utilized siRNA sequences were as follows: Si-CXCL10, 5′-GCCTTATCTTTCTGACTCTAA-3′; Si-NC, 5′-CCTAAGGTTAAGTCGCCCTCG-3′.

RNA extraction and quantitative reverse transcription (RT-qPCR)

Total RNA from KGN cells was extracted using TRIzol® Reagent (Thermo Fisher Scientific, Waltham, MA, USA) following the manufacturer’s instructions. DNA was digested using Dnase I (18068015; Thermo Fisher Scientific, Waltham, MA, USA) at 25 °C for 1 h. Contamination assessment was performed using a 20-minute agarose gel electrophoresis at 130 V. The quality of the RNA was evaluated using NanoDrop One (Thermo Fisher Scientific, Waltham, MA, USA), targeting an A260/280 ratio of 1.8–2.0. RNA integrity was evaluated using an Agilent 2100 bioanalyzer, with an RNA integrity number (RIN) ranging from 7 to 10. The SPUD assay was used to test for inhibition, according to the SPUD Assay for Detection of Assay Inhibitors Protocol (Merck, Rahway, NJ, USA). For each microgram of total RNA, 0.5 µg of primer or primer-adaptor was added to 2 µg of total RNA in a sterile Rnase-free microcentrifuge tube.

The mixture, in a total volume of ≤15 µL of water, was heated to 70 °C for 5 min.

Reverse transcription of cDNA was performed using 200 units of M-MLV reverse transcriptase (M170A; Promega, Madison, WI, USA) in a 25 μL volume, incubating for 60 min at 37 °C with random primers or 42 °C with other primers or primer-adaptors. RT-qPCR was carried out on ABI7500 Real-Time PCR Detection System with a 20 μL reaction system, including 2 μL of cDNA and AceQ Universal SYBR qPCR Master Mix (Q511; Vazyme, Nanjing, China) containing 50 μmol/L dNTP, 1.5 mmol/L Mg2+, 20 U AceTaq DNA polymerase according to the protocol. The PCR process comprised an initial pre-denaturing step at 95 °C for 2 min, followed by 42 cycles of 95 °C for 30 s, 58 °C for 30 s, and 72 °C for 30 s.

The dissolution curve must be unimodal. Standard curves with slope and y-intercept were developed, aiming for an r2 value of 0.99. The dynamic range of the PCR reaction should exhibit linearity, with a Cq variation at the lower limit of 10. The limit of detection (LOD) was determined to be 2.5 target molecules with a 95% confidence interval. The 2−ΔΔCt method was used to determine the relative expression of CXCL10 (NM_001565.4), normalized to GAPDH (NM_002046.7), using the Applied Biosystems ViiA™ 7 Real-Time PCR System, with GAPDH serving as control. The primer sequences used for GAPDH and CXCL10 were as follows: 5′-GTGGCATTCAAGGAGTACCTC-3′, 5′-TGATGGCCTTCGATTCTGGATT-3′, 5′-AAGTATGACAACAGCCTCAAG-3′ and 5′-TCCACGATACCAAAGTTGTC-3′.

The amplicon length was approximately 100 bp, corresponding to the gene’s exon, which was verified in NCBI. The comparative 2−ΔΔCt method assessed the stability of each gene by determining the standard deviation of Cq differences. No Cq value was detected for the No Template Control (NTC). Outliers, defined as values significantly deviating from the other two values within three technical replicates were excluded from statistical analysis. Each experiment involved three biological and technical replicates.

Western blot

Cells were suspended in CellScope’s RIPA lysis buffer, and protein concentration was determined using the bicinchoninic acid assay (Thermo Fisher Scientific, Waltham, MA, USA). A 40 μg protein sample was loaded on a 10% gel for sodium dodecyl polyacrylamide gel electrophoresis (SDS-PAGE). The proteins were transferred to a polyvinylidene fluoride (PVDF) membrane which was blocked using 5% skim milk and subsequently incubated with the following antibodies: anti-GAPDH (ab9484; Abcam, Cambridge, UK), anti-CXCL10 (ab306587; Abcam, Cambridge, UK), anti-peroxisome proliferator-activated receptor (PPAR) β (ab310323; Abcam, Cambridge, UK), anti-long-chain acyl-coenzyme A synthetases 1 (ACSL1, ab177958; Abcam, Cambridge, UK), anti-caspase (ab32351; Abcam, Cambridge, UK), anti-B-cell lymphoma 2 (Bcl-2, ab182858; Abcam, Cambridge, UK), and anti-Bcl-2-associated X protein (Bax, ab32503; Abcam, Cambridge, UK). The blots were then incubated with a secondary goat anti-rabbit IgG (H+L) antibody (A16104; Thermo Fisher, Waltham, MA, USA) for 1 h at room temperature. Immunoreactive bands were visualized using an enhanced chemiluminescence reaction (Pierce, Appleton, WI, USA) following standard protocols.

Annexin V-fluorescein isothiocyanate (FITC)/propidium iodide (PI) assay

KGN cells were placed in a six-well culture dish and cultured in an incubator at 37 °C with 5% CO2. After 48 h of culture, cells from each group were then treated with 5 μL of annexin V-FITC and 5 μL of PI solutions (E-CK-A211; Elabscience, Houston, TX, USA) at 37 °C for 20 min in the absence of light. Apoptotic cells were counted using a flow cytometer (BD Biosciences Co. Ltd, Franklin Lakes, NJ, USA).

Enzyme-linked immunosorbent assay (ELISA)

Human estradiol (E2) levels were detected using human estradiol ELISA kit (MM-0777H1; Jiangsu Meimian Industrial Co., Ltd., Jiangsu, China) according to the instructions of the manufacturer. The absorbance was measured at 450 nm using a microplate reader.

Statistical analysis

We performed statistical analysis using SPSS 21.0 software (SPSS Inc., Chicago, IL, USA). Data are expressed as mean ± standard deviation (SD). To assess differences between two or more groups, we employed an unpaired Student’s t-test or χ2-test, and a one-way analysis of variance, respectively. Tukey’s multiple comparison test was applied when necessary. Each experiment was conducted with a minimum of three biological replicates, and statistical significance was defined as p-levels < 0.05.

Results

Expression of CXCL10 in POF

The PCA analysis revealed that the expression of CXCL10 was higher in the POF group than in the control group (Fig. 1A). Additionally, the violin plot demonstrated a significant increase in CXCL10 expression levels in the POF group compared to the control group (Fig. 1B). ROC curve analysis using PlotROC revealed that CXCL10 had a relatively accurate diagnostic potential for POF with an AUC value of 1 (Fig. 1C). Overall, these results indicate that the expression of CXCL10 is dysregulated in POF.

Figure 1 The expression of CXCL10 in polycystic ovarian failure (POF) is dysregulated.

(A) Principal component analysis (PCA) expression. Larger points on the diagram represent higher expression values. In PCA, data dimensionality of the data set was reduced, allowing the creation of a two-dimensional scatter plot for an initial sample distribution assessment. (B) ViolinPlot of the transcription levels of CXCL10 in POF and control; (C) receiver operating characteristic (ROC) curve analysis of CCXCL10.

Global regulation of CXCL10 in POF

To understand the global regulation of CXCL10 in POF, we conducted a differential expression analysis to identify DEGs in the control group of POF using Debylimma.

This analysis revealed 3,362 up-regulated and 3,969 down-regulated DEGs (Fig. 2A, Table S1). Furthermore, we found 1,304 up-regulated and 1,315 down-regulated DEGs between the high and low CXCL10 expression groups of POF (Fig. 2B, Table S2). The overlap of DEGs between the POF-control group and the high- and low-CXCL10 expression groups was determined using PlotVenn, as shown in Fig. 2C. A total of 786 DEGs, uniformly up-regulated or down-regulated, were identified as POF-related genes dysregulated by CXCL10. These genes were further analyzed using QuadrantPlot (Fig. 2D).

Figure 2 Global regulation of Cxcl10 in polycystic ovarian failure (POF).

(A) Differentially expressed genes (DEGs) in the POF control group; (B) Volcano map of the DEGs in the high- and low-expression POF groups. Larger absolute values of logFC indicate greater differences in the multiple of gene expression between sample groups, while −log10P.adjust indicate the significance of expression differences. The larger the ordinate, the more significant the difference expression; (C) overlap between DEGs in the POF-control group and the high- and low- CXCL10 expression group using Plot Venn; (D) CXCL10-dysregulated genes in POF assessed using QuadrantPlot; blue represents DEGs with high expression (logFC > 0) in POF-Control and high expression (logFC > 0) in high- and low-expression of CXCL10. Red represents high expression (logFC > 0) in the POF-Control and low expression in in high- and low-expression of CXCL10 (logFC < 0). Brown-yellow represents low expression in POF-Control (logFC < 0) and low expression in CXCL10 (logFC < 0). Dark red represents low expression in POF-Control (logFC < 0) and high to low expression in CXCL10 (logFC > 0).

Enrichment analysis

To explore the potential BP and pathways affected by the CXCL10 gene in POF progression, we conducted GO and KEGG analyses on DEGs using the RunMulti Group cluster Profiler. GO enrichment analysis revealed significant involvement of these DEGs in cell chemotaxis, negative regulation of response to DNA damage stimulus, and diadenosine polyphosphate metabolic processes (Fig. 3A, Table S3). KEGG analysis showed significant enrichment of these genes in the PPAR signaling pathway, adiponectin signaling pathway, drug metabolism-cytochrome P450, and metabolism of xenobiotics by cytochrome P450 (Fig. 3B, Table S3). Gene Set Enrichment Analysis (GSEA) indicated strong activation of the PPAR signaling pathway in POF (Fig. 3C, FDR = 0.007 and enrichment score = 0.813). The map generated using RunPathview displayed significant changes in genes related to the PPAR signaling pathway, including LXRa, ACS, CYP4A1, ACO, CPT-1, MCAD, POAR, and ADIPO. Among these, ACS was the most upregulated in POF (Fig. 4, Table S2). These results indicated that CXCL10 activates the PPAR signaling pathway in POF.

Figure 3 Enrichment analysis.

(A and B) Gene ontology and kyoto encyclopedia of genes and genomes analysis on differentially expressed genes was used to explore the potential biological processes and pathways affected by CXCL10 gene involved in polycystic ovarian failure progression using RunMulti group cluster profiler; (C) Gene set enrichment analysis was performed using RunGSEA.

Figure 4 Changes in the peroxisome proliferator–activated receptor signaling pathway-related genes, including LXRa, ACS, CYP4A1, ACO, CPT-1, MCAD, POAR, and ADIPO, assessed using RunPathview.

Diagnostic efficacy of CXCL10-based clinical model

ROC curves for CXCL10-dysregulated genes in POF were analyzed using PlotROC. RunLASSO performed characteristic gene selection for these genes, identifying three feature genes, namely Cxc110, Raf1, and ITGA2, with non-zero regression coefficients (lambda.min = 0.0006). The Lambda diagram (Fig. 5A) illustrates the diagnostic performance of the CXCL10-based model across different Lambda values, with the best Lambda at 0.001 and the minimum at 0.0006. The LASSO model diagram (Fig. 5B) indicates the model’s confidence in predicting POF diagnosis. Finally, the ROC curve (Fig. 5C) demonstrates that the CXCL10-based model possesses excellent diagnostic efficiency (AUC = 1) for POF in the training set, serving as a potential diagnostic biomarker for POF.

Figure 5 Diagnostic efficacy of the CXCL10-based clinical model.

(A) Lambda diagram of the diagnostic efficiency of CXCL10-dysregulated genes in POF under different lambdas; (B) The LASSO model diagram of the diagnostic efficiency of CXCL10-dysregulated genes in POF under different log(Lambda); (C) Receiver operating characteristic (ROC) curve of CXCL10-dysregulated genes in POF were analyzed using PlotROC to assess the CXCL10-based model’s performance. Subsequently, RunLASSO was used to select characteristic genes dysregulated in POF through CXCL10 and possessing POF diagnostic efficiency; (D) Box plot of the transcriptional expression levels of model genes, including CXCL10, ITGA2, and RAF1.

Moreover, the expression of the model genes significantly differed between the POF and control groups. Additional ROC curves (Fig. S1) further support the diagnostic potential of CXCL10-based model genes. The CXCL10 expression was higher in the case group than in the control group, indicating its association with POF. Similarly, lower Itga2 and Raf1 expression in the case group suggests their potential as diagnostic markers for POF (Fig. 5D).

CXCL10 affects the progression of POF by influencing the PPAR signaling pathway

To verify the role of CXCL10 in POF, we constructed a DDP-induced KGN cell model of injury, silencing the expression of CXCL10 in these cells. Firstly, we determined that DDP at concentrations exceeding 10 μg/mL significantly reduced cell proliferation at both 24 and 48 h (Fig. 6A), indicating that this concentration of DDP can induce KGN cell damage.

Figure 6 CXCL10 affects the secretion of follicle-stimulating hormone and estradiol in KGN POF model cells.

(A) Assessment of cisplatin concentration needed to induce KGN cell; (B) silencing efficiency of CXCL10 SiRNA in KGN cells assessed using reverse transcription quantitative polymerase chain reaction (RT-qPCR); (C and D) Expression of CXCL10 in POF cell model before and after CXCL10 silencing assessed using RT-qPCR and Western blot; the concentration of DDP that was finally used was 10 μg/mL for 48 h. (E) Secretion of estradiol (E2) was detected in KGN cells before and after CXCL10 silencing. **p < 0.01, ***p < 0.001, ****p < 0.0001.

Si-CXCL10#2 was chosen for further experiments due to its efficacy in silencing CXCL10 in KGN cells, as confirmed by RT-qPCR analysis (Fig. 6B). RT-qPCR and WB analyses revealed increased expression of CXCL10 in the POF cell model group compared to the control group. However, CXCL10 expression was significantly reduced in the POF+Si-CXCL10 group compared to the si-NC group (Figs. 6C and 6D). Estradiol secretion was significantly inhibited in the KGN cell model of injury after 48 h of culture, but this inhibition was alleviated upon interference with CXCL10 expression (Fig. 6E). These findings demonstrated that CXCL10 interference had a protective effect against DDP-induced granulosa cell damage, preserving granulosa cell function.

Subsequently, we examined the effect of CXCL10 on apoptosis. The results indicated a substantial increase in the apoptosis rate of KGN cells in the POF group compared to the control group. However, silencing CXCL10 resulted in significant inhibition of apoptosis when compared with the si-NC group (Fig. 7A). This was further corroborated by WB analysis, which revealed that the expressions of pro-apoptotic genes Bax and caspase increased while the anti-apoptotic gene Bcl-2 decreased in the POF group compared to the control group. Compared to the Si-NC group, the expressions of Bax and caspase were decreased in the Si-CXCL10 group, and the expression of Bcl-2 was significantly increased (Fig. 7B).

Figure 7 CXCL10 affects the apoptosis and the peroxisome proliferator–activated receptor (PPAR) signaling pathway in KGN cell model of polycystic ovarian failure (POF).

(A) Apoptosis of KGN cells following CXCL10 inhibition assessed using flow cytometry; (B) expressions of caspase, anti-B-cell lymphoma 2 (Bcl-2), and anti-B-cell lymphoma 2 (Bcl-2) detected by Western blot in KGN cells following CXCL10 inhibition; (C) expressions of peroxisome proliferator–activated receptor (PPAR)β and long-chain acyl-coenzyme A synthetases 1 in the PPAR signaling pathway in KGN cells following CXCL10 inhibition detected by Western blot; *p < 0.05, **p < 0.01, ***p < 0.001, ****p < 0.0001.

In addition, we examined the effect of CXCL10 on the PPAR signaling pathway, revealing a significant increase in the expressions of PPARβ and ACSL1 in the POF group compared to the control group. However, CXCL10 silencing led to decreased levels of PPARβ and ACSL1 compared to the Si-NC group (Fig. 7C), indicating that CXCL10 impacts the PPAR signaling pathway.

Discussion

In this study, we developed a CXCL10-based gene cluster model as a potential diagnostic biomarker for POF, a multifactorial and heterogeneous disease characterized by amenorrhea, decreased estrogen levels, and increased female gonadotropin levels (Komorowska, 2016). The incidence of POF is increasing annually. Furthermore, POF is a major cause of infertility in women of childbearing age, yet its etiology remains complex and poorly understood (Bai & Wang, 2022). While POF diagnosis typically relies on a combination of clinical evaluation, hormone testing, and imaging studies, there is no single definitive diagnostic marker.

The diagnosis of POF typically involves a combination of clinical evaluation, hormone testing, and imaging studies. Various tests, such as assessing the levels of FSH, LH, estradiol, anti-mullerian hormone (AMH), and anti-ovarian antibodies (de Kat, Broekmans & Lambalk, 2021; Szeliga et al., 2021) can assist in diagnosis. However, genetic markers for the diagnosis of POF are currently lacking. In our study, we explored CXCL10, a chemokine involved in inflammatory and immune responses, as a potential diagnostic marker (Reschke & Gajewski, 2022).

While limited research has explored the role of CXCL10 in POF, our study reveals that its expression is dysregulated in individuals with this pathology. Dysregulation of CXCL10 implies abnormal expression in patients with POF compared to healthy individuals. We also demonstrated that CXCL10 is involved in the regulation of progesterone and estradiol secretion, and the apoptosis of KGN cells in a POF cell model. In addition, we confirmed the high diagnostic potential of CXCL10 for POF through the AUC value of 1, which indicates perfect accuracy for diagnosing the condition.

Although these findings suggest a potential association between CXCL10 and POF, the exact mechanisms behind the contribution of CXCL10 to the development of POF remain unclear. POF is a complex condition influenced by various genetic, hormonal, and immune factors (Chapman, Cree & Shelling, 2015; Shamilova et al., 2013) CXCL10 represents just a component of this complex interplay, and further comprehensive research is needed to fully understand its significance in POF. In the present study, we demonstrated that the CXCL10-based model, including CXCL10, ITGA2, and RAF1, exhibits promise as a potential diagnostic biomarker for POF. ITGA2 encodes the alpha-2 subunit of integrin receptors known for their essential roles in cell adhesion, migration, signaling, and interaction with the extracellular matrix (Chuang et al., 2018; Ding et al., 2015). ITGA2 participates in various physiological processes, including platelet function, immune cell adhesion, and cell signaling.

Dysregulation of ITGA2 contributes to bleeding disorders, cancer metastasis, fibrosis, and inflammatory conditions (Wu et al., 2014). RAF1 is a gene encoding the Raf-1 protein, a key component of the MAPK signaling pathway that regulates cellular processes such as proliferation, differentiation, and survival (Iglesias-Martinez et al., 2023). Mutations or overactivation of RAF1 can contribute to genetic disorders and cancer (Bekele et al., 2021; Simanshu, Nissley & McCormick, 2017), highlighting its importance in normal cellular function and disease development. Although the MAPK signaling pathway plays a crucial role in POF, the specific function of RAF1 in POF remains unclear (Liu et al., 2021; Mantawy, Said & Abdel-Aziz, 2019). These genes have yet to be comprehensively characterized, and collectively, they may provide valuable information for accurately diagnosing POF when their expression levels are analyzed. In summary, the dysregulation of CXCL10 is observed in POF, accompanied by DEGs and the activation of the PPAR signaling pathway. The CXCL10-based model, including other genes such as Itga2 and Raf1, may serve as a potential diagnostic biomarker for POF. However, these genetic markers are not present in all cases of POF, and genetic testing may not provide a conclusive diagnosis for every individual with POF.

The PPAR signaling pathway is a crucial regulatory pathway involved in the control of various physiological processes, including lipid metabolism, glucose homeostasis, inflammation, and cell differentiation (Wang, Dougherty & Danner, 2016). The pathway is controlled by a group of nuclear receptor proteins known as PPARs, which act as ligand-activated transcription factors. There are three subtypes of PPARs: PPAR-alpha, -delta ( or beta), and -gamma, each with distinct tissue expression patterns and physiological roles (Wagner & Wagner, 2020). In the present study, significant changes were observed in the PPAR signaling pathway-related genes, including LXRa, ACS, CYP4A1, ACO, CPT-1, MCAD, POAR, and ADIPO, with ACS being the most upregulated in POF. Studies suggest that the PPAR signaling pathway might play a crucial role in the progression of POF.

Transcriptome analysis suggests that cyclophosphamide hampers cholesterol biosynthesis, causing premature ovarian failure. Furthermore, it reveals the importance of the PPAR signaling pathway and ovarian infertility genes in the progression of POF (Li et al., 2019). Said et al. (2016) demonstrated that resveratrol enhanced ovarian function by increasing AMH levels and reducing ovarian inflammation. This was mainly achieved by upregulating the expression of PPAR-γ and SIRT1, subsequently inhibiting the production of NF-κB-induced inflammatory cytokines. Ju et al. (2023) demonstrated that reducing the expression of PPAR-γ modifies the protein expression of the primary target and increases estrogen levels. In this study, enrichment analysis and Western blot confirmed the activation of the PPAR signaling pathway induced by CXCL10 in POF. These findings suggest that dysregulation of CXCL10 in POF may influence these processes through the activation of the PPAR signaling pathway. Understanding how CXCL10 and the PPAR signaling pathway interact in POF can provide valuable insights into the molecular basis of the condition and potentially lead to the development of targeted therapeutic strategies.

The CXCL10-based gene cluster model for POF may have certain limitations as a diagnostic biomarker. POF is a complex condition with diverse underlying causes, including genetic factors, autoimmune disorders, and chemotherapy. The CXCL10-based gene cluster model might not encompass the full range of genetic and molecular variations associated with POF, limiting its effectiveness in detecting other causes-induced POF. Adequate sample sizes of healthy and individuals with POF are essential for accurate model development and validation. The diagnostic accuracy of a biomarker can vary among different populations due to genetic and environmental factors. Thus, it is critical to validate the CXCL10-based gene cluster model across diverse populations. These limitations are not specific to the CXCL10-based gene cluster model alone but to the development of any diagnostic biomarker, emphasizing the need for further research, validation, and collaboration among researchers to enhance diagnostic accuracy and clinical utility for POF.

Conclusion

In summary, the dysregulation of CXCL10, along with associated DEGs and PPAR signaling pathway activation, is associated with the progression of POF. Subsequently, combining CXCL10 with other genes such as Itga2 and Raf1, could serve as a potential diagnostic biomarker for POF. Further research in this area can enhance our understanding of the mechanisms underlying POF, potentially improving diagnostics and therapeutic strategies.

Supplemental Information

Supplemental Information 1 Raw data.

Click here for additional data file.

Supplemental Information 2 Supplemental Files.

Click here for additional data file.

Supplemental Information 3 MIQE Checklist.

Click here for additional data file.

Additional Information and Declarations

Competing Interests

Author Contributions

Data Availability

The authors declare that they have no competing interests.

Ying Qin conceived and designed the experiments, performed the experiments, analyzed the data, authored or reviewed drafts of the article, and approved the final draft.

Canliang Wen analyzed the data, authored or reviewed drafts of the article, and approved the final draft.

Huijiao Wu analyzed the data, prepared figures and/or tables, and approved the final draft.

The following information was supplied regarding data availability:

The raw measurements are available in the Supplemental Files.

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
