# Peer review of "CXCL10-based gene cluster model serves as a potential diagnostic biomarker for premature ovarian failure"

_PeerJ, doi:10.7717/peerj.16659_

## Round 0.1 · original submission · Major Revisions

Major issues:
1. The Results section in the Abstract was poorly written, because the results from cell experiments were not reflected in this section.
2. How to define the POF-control group and the High-Low group? Please provide the details of the grouping methodology and criteria.
3. Line 234: [ACS showed the most 235 upregulated in POF]. Please show the data supporting this result.
4. [DDP-induced KGN cell injury model] should be corroborated by previous literature.
5. Figure 6C: The concentration of DDP that was finally used here should be clearly demonstrated.

Minor issues:
Professional English editing is required to improve the language quality of this work.
For instance:
1. [Differential Expressed Genes (DEGs) in the control group of POF] means what?
2. [POF-control group and the High-Low group DEGs of Cxcl10] means what?
3. [affected by the Cxcl10 gene, using RunMulti Group cluster Profiler]; [RunLASSO performed characteristic gene selection for POF dysregulation genes affected by Cxcl10, which demonstrated diagnostic efficacy for POF]?; [was silted]?; [was add ]?; [which incubated]?; [was blast]?; [results demonstrates]?; [that used]?; [CXCL10 involved]?; [have been proved]?; [Several studies were also should that]?; [also indicate]? These should be completely rewritten.

**Language Note:** The Academic Editor has identified that the English language must be improved. PeerJ can provide language editing services - please contact us at [email protected] for pricing (be sure to provide your manuscript number and title). Alternatively, you should make your own arrangements to improve the language quality and provide details in your response letter. – PeerJ Staff

**ADDITIONAL STAFF NOTE:** The authors state that they measure FSH secretion by KGN cells. However, FSH has not been reported to be secreted by these cells before. Maybe authors meant FSH-receptor instead?

Reviewer 1 ·

Basic reporting

In this manuscript, the authors have presented the Cxcl10-based gene cluster as a marker for POF. Based on the presented work, I have several global concerns.

First, many of the methods are unclear. The abstract states that PCA was performed on POF patients from the GEO database. However, the methods state that the data collected was from three POF mouse lines. Further, if Cxcl10 is only one gene, why is a PCA analysis needed to compare the expression leveled for the cases and controls. If mouse lines are being used, what “POF mouse lines” are being used. If human patients are being used, how is POF being define?

For Figure 2, they then divide DEGs based on low and high Cxcl10 expressing genes. Again, if this is only one gene, where is this variation coming from. I was unable to get much further than this in the paper because the methods and experiments being performed were just unclear.

Overall, the paper is in need of many additional details to describe the logic behind the experiments being performed. There are plenty of facts presented for the individual genes that are described in the paper; however, there is not a complete description as to what experiments are being done or why they are being done.

Experimental design

Covered above.

Validity of the findings

In the abstract, they state that given their AUC of 1, this indicates a perfect accuracy for diagnosing POF. Clearly, this cannot be the case, and the necessary data to demonstrate this has not been collected.

Additional comments

The authors should consider updating their terminology from POF to POI unless they have a specific reason for using this wording.

Reviewer 2 ·

Basic reporting

1.1 This study constructed a diagnostic model using the transcriptome data of Premature ovarian failure samples, and designed a wet test to verify the bioinformatics results. In general, the author's research intention is clear, but the writing level of the manuscript needs to be greatly improved in order to better benefit the readers. It should be noted that the writing and English grammar of the article need to be carefully proofread throughout, and it is best done by professional organizations or fluent English speakers in the same field. For example, the first letter of the sentence in line 42 is not capitalized, and such details need to be proofread one by one.
1.2 References are selected and cited appropriately.
1.3 The structure of the article is clear, and the charts and original data are in line with the specification.

Experimental design

2.1 During bioinformatics analysis, the selection of sample cohort has a huge impact on the results. There are two questions in the selection of data set for this study.During analysis, large samples are necessary. This study only used 6 samples. What is the reason for choosing GSE39501?
2.2 What is High-Low group? There is no basis for this grouping in the materials and methods. And since there are DEGs, differential expression analysis should have been performed, which was also missed in the materials and methods.
2.3 Is it reasonable to just use a Venn diagram to determine the DEGs affected by Cxcl10? Is it more reasonable to use pearson correlation analysis or sperman correlation to determine the DEGs affected by Cxcl10?
2.4 GEO’s website should be explained in detail.
2.5 In the materials and methods, the basic reagent origin brand needs to be listed in detail, such as WB secondary antibody.

Validity of the findings

3. The research gene selected for this study is Cxcl10. What is the scientific basis for initially locating Cxcl10 in POF.
3.2.The results show that the PPAR pathway is significantly activated in POF. What are the FDR and ES in Figure 3C?
3.3 In the results of Figure 5A-B, the Lambda when Itga2 and Raf1 are identified should be explained in detail.

Reviewer 3 ·

Basic reporting

1.1 Premature ovarian failure (POF) is the term usually used to describe women aged younger than 40 years, who present with amenorrhoea, hypergonadotropic hypogonadism, and infertility. POF has serious health consequences, including psychological distress, infertility, osteoporosis, autoimmune disorders, ischaemic heart disease, and increased risk of mortality. In this study, the authors gene CXCL10 established a gene cluster model, which was able to predict the prognosis of POF patients well. In addition, this study demonstrated that CXCL10 involved in the regulation of progesterone and estradiol secretion, and the apoptosis of KGN cells in a POF cell model. Overall, this is a comprehensive study. However, Authentic English expressions and grammatical norms will help readers to understand the research content correctly, Such as line 25 have a grammar error.
1.2 The references are generally appropriate, but some parts need to be appropriately supplemented with necessary references, such as line 314, “It is worth noting that POF is a complex condition influenced by various genetic, hormonal, and immune factors.” this requires the identification of specific reference sources.
1.3 There are no obvious errors and omissions in the structure, diagrams and data of the manuscript, which basically meet the requirements for publication.

Experimental design

2.1 The CXCL10 serves as the key to this study and its background needs to be reflected in the ABSTRACT section.
2.2 In the ABSTRACT section, the full name of the acronym should be added where it occurs, such as GEO, etc.
2.3 For relevant clinical studies of CXCL10 in POF, this should be added to the Introduction.
2.4 In the second paragraph of the INTRODUCTION section, a description of relevant studies in which gene expression signatures predicted the prognosis of patients with POF should be provided.
2.5 In line 88, for data sources obtained from the GEO database, authors should clearly indicate their specific URL. In addition, the screening conditions and specific information about the sample need to be indicated in detail.
2.6 In lines 93-94, it is suggested that a relevant reference for PCA be provided.

Validity of the findings

3.1 In line 24, the authors should specify what is meant by "control group".
3.2 In line 45, please simplify the "Conclusion" section to highlight the main findings of the study.
3.3 In line 83-84, the author should provide a relevant description of the main work of this study to be able to enable the reader to understand the general content of the study by being able to understand it in the introduction.
3.4 The Lambda for identifying Itga2 and Raf1 should be explained in detail, as shown in the results of Figure 5A-B.

Additional comments

Authors should check the full text for "Cxcl10" or "CXCL10". Attention needs to be paid to the specific use of the term.

---

## Round 0.2 · accepted · Accept

My concerns have been well addressed. I think this revised version could be considered for publication in this journal, except for the following issue, which should be addressed at the proof stage.

As requested, the authors added the following literature on the drug-induced POF model to their rebuttal letter, but they forgot to add it to the revised version of the main text:

Wang R, Wang L, Wang L, Cui Z, Cheng F, Wang W, Yang X. FGF2 Is Protective Towards Cisplatin-Induced KGN Cell Toxicity by Promoting FTO Expression and Autophagy. Front Endocrinol (Lausanne). 2022 Jun 16;13:890623.

Therefore, the authors should show this reference in line 155 after [For cisplatin (DDP) treatment].

Reviewer 2 ·

Basic reporting

The authors have largely addressed my concerns and the revised version is acceptable.

Experimental design

no comment

Validity of the findings

no comment

Reviewer 3 ·

Basic reporting

The author's revisions and responses are effective and generally improve the quality of the manuscript.

Experimental design

The author made the necessary revisions.

Validity of the findings

The author made the necessary revisions.

Additional comments

no comment